# Removal Capacity and Mechanism of Modified Chitosan for Ochratoxin A Based on Rapid Magnetic Separation Technology

**DOI:** 10.3390/foods14040666

**Published:** 2025-02-15

**Authors:** Xueyan Xin, Mina Nan, Yang Bi, Huali Xue, Liang Lyu, Daiwei Jiang, Hongjuan Chen, Qifang Luo

**Affiliations:** 1College of Science, Gansu Agricultural University, Lanzhou 730070, China; xinxy@st.edu.cn (X.X.); xuehl@gsau.edu.cn (H.X.); jiangdw@st.edu.cn (D.J.); chenhj@st.edu.cn (H.C.); luoqf@st.edu.cn (Q.L.); 2Basic Experiment Teaching Center, Gansu Agricultural University, Lanzhou 730070, China; 3College of Food Science and Engineering, Gansu Agricultural University, Lanzhou 730070, China; biy@gsau.edu.cn; 4College of Biological and Pharmaceutical Engineering, Lanzhou Jiaotong University, Lanzhou 730070, China; lyuliang@mail.lzjtu.cn

**Keywords:** ochratoxin A, adsorption, nano Fe_3_O_4_ modified chitosan, wine, mechanisms

## Abstract

Ochratoxin A (OTA) exposure in food is very dangerous to human health. Therefore, the development of a fast and efficient technique for OTA removal has become an urgent research topic in the field of food safety. Nano Fe_3_O_4_ modified chitosan nanocomposite (nano-Fe_3_O_4_@CTS) was synthesized as a rapidly separable and safe adsorbent and was used to adsorb OTA in wine. FT-IR, XRD, and VSM characterization methods indicated that chitosan was successfully modified by Fe_3_O_4_ and exhibited good magnetism. The adsorption and kinetics isotherms between OTA and nano-Fe_3_O_4_@CTS were studied by the Langmuir equation and the pseudo-second order kinetics equation. The mechanism of OTA adsorption on nano-Fe_3_O_4_@CTS nanoparticles was the combined effect of physical adsorption and chemisorption. The negative *ΔH°*, *ΔG°* and *ΔS°* values proved that the adsorption was a spontaneous and exothermic process. Nano-Fe_3_O_4_@CTS with a high maximum adsorption capacity of 5018.07 ng/g at 25 °C can rapidly separate the matrix immobilized OTA from wine, and to a certain extent retains some of the wine quality after OTA removal.

## 1. Introduction

The principal producers of ochratoxin A (OTA) are *Penicillium* and *Aspergillus* genera [1]. A large amount of research has reported the harmful consequences of OTA on human health, including mutagenic, teratogenic, carcinogenic, hepatotoxic and neurotoxic effects, and the International Agency for Research on Cancer (IARC) classified OTA as a Group 2B carcinogen [2,3]. OTA has a wide range of contamination and can contaminate a variety of foods, such as corn, wheat, coffee, wine, juice, nuts, and animal feed [4]. As the second largest source of OTA intake (second only to cereals), the highest OTA residue level in wine formulated by Regulation 2022/1370 of the European Commission was 2.0 µg/L [5]. Kochman et al. [6] analyzed the content of OTA in Polish, Spanish, and French red wines, and their findings indicated that the OTA content in all red wines was higher than the upper limit, with values of 6.629 µg/L, 6.848 µg/L, 6.711 µg/L, respectively. As a result, it is particularly important to rigorously regulate and lower the amount of OTA in foods. However, OTA is a weak organic acid (pKa = 4.4 and 7.3), which can withstand high temperatures and high acidity [6]. OTA is often difficult to remove from contaminated food due to its strong chemical and thermal stability [7].

Removal can effectively control the pollution of OTA in foods, thereby reducing the harm to humans and animals. At present, the known removal methods mainly include physical, chemical, and biological methods. Chemical methods can effectively remove mycotoxin in foods through using strong oxidants, acids, alkalis, and other chemicals, which convert OTA into other substances by chemical reaction and destroy the structure of the mycotoxin [4]. Jalili et al. [8] studied the removal of OTA in pepper by alkali ammonia, sodium bicarbonate, sodium hydroxide, potassium hydroxide, and calcium hydroxide, respectively. The results showed that there was no significant difference in the removal effect of the five compounds, and the highest sodium hydroxide was more than 50%. However, when OTA are removed by chemical methods, residual chemical reagents may have an impact on human health. In recent years, biological methods using microorganisms to remove and degrade mycotoxin have been identified as an effective and potential method. The mechanism is to destroy the chemical structure of the mycotoxin by microbiology, or the enzymes secreted by microbiology [4]. Nora et al. [9] found that commercial peroxidases could reduce OTA content by 41% in grape juice within 5 h. During the production of wine and grape juice, Saccharomyces cerevisiae and lactic acid bacteria can effectively control the concentration of OTA [10]. Biological methods usually require specific microorganisms and enzymes, which often lead to higher production cost resources. In addition, since the efficiency of biological methods may not be as high as that of physical or chemical methods, it requires longer time and greater investment. Physical adsorption methods manifest promising prospects in removing mycotoxins, such as heat treatment, radiation, adsorption, etc., but few of these physical removal materials have practical applications [11]. Among them, the adsorbent has drawn the interest of numerous researchers because of its simple application and low cost. The application of adsorbents seems to be the most common way to remove OTA [12]. For example, bentonite was used to adsorb OTA, but the adsorption rate (8%) was fairly low [13]. Rotter et al. [14] used carbon to remove OTA and the outcomes demonstrated that the OTA removal rate could approach 90% but with low separation efficiency. Up to now, the reported adsorbents are relatively single, low adsorption rates, and have low separation efficiency. Thus, it is essential to develop practical, safe, easily separable adsorbents for the removal of mycotoxins.

Chitosan is a natural polysaccharide mainly derived from chitin and is obtained by partial deacetylation of chitin under alkaline conditions. The main sources of industrial-scale production of chitosan are marine crustaceans, the shells of shrimps and crabs and the bone plates of squid [15]. Chitosan has gained popularity for its remarkable properties, such as innocuous, high adsorption and affinity, biocompatible, biodegradable, antimicrobial, environment-friendly, and low cost [12]. In addition, the amino and hydroxyl groups in chitosan can be used as active sites and modified, making it widely used in different industries, such as pharmaceutical, food, agriculture, cosmetics, drug delivery, biotechnology, and biomedicine. Since 2011, the EU has approved the use of chitosan for the removal of contaminants in wine, such as OTA. Bornet et al. [16] suggested that chitosan as an adsorbent (2 and 5 g/L) can significantly reduce the concentration of OTA in red (56.7–83.4%), white (26.1–43.5%) and sweet wine (53.4–64.5%) The study also showed that chitosan can be used to clarify and eliminate OTA and other pollutants, such as metal ions. However, because of its viscosity, chitosan usually forms an emulsion in solution, which is difficult to separate in a solution by traditional filtration, centrifugation, and sedimentation [17]. Tradition physical adsorption methods are limited by the time-consuming operation and hard removal of the matrix-immobilized OTA [18]. To solve this problem, the Magnetite modification method was applied to achieve the purpose of convenient and rapid recovery of the adsorbent. As one of the most common magnetic materials, Fe_3_O_4_ has broad application prospects in many fields [19]. Magnetically modified chitosan shows extensive promising prospects in the field of wastewater treatment. Omidinasab et al. [20] developed Fe_3_O_4_ modified chitosan material to efficiently adsorb vanadium (V), and its adsorption is 186.6 mg/g, under optimized conditions. A novel magnetic graphene oxide modified with chitosan (MGO-CTS) was synthesized as an adsorbent aimed at the removal of mycotoxins OTA, ZEN, AFB_1_ and can be reduced to mycotoxins at 50 °C and pH 5 [15]. However, there is a little research on its application as an adsorbent to eliminate OTA in complex samples, and the approach will be a fast, efficient, and secure OTA removal method.

The study aims to evaluate the effectiveness of the prepared adsorbent (nano Fe_3_O_4_@CTS) in reducing the OTA content in wine. The research extensively examined the optimization parameters for nano-Fe_3_O_4_@CTS regarding OTA reduction, including adsorbent dosages, adsorption time, pH, and temperature. The adsorption mechanism was evaluated according to thermodynamics and kinetics equations. The nano-Fe_3_O_4_@CTS nanoparticle was applied to eliminate OTA in wine, and the practical application was studied through the changes in wine chemical parameters.

## 2. Materials and Methods

### 2.1. Chemicals and Reagents

The OTA standard (C_20_H_18_CLNO_6_, Certified value: 99%, 1 mg) and PriboFast IAC-040-3ZY were both purchased from Pribolab Biological Technical Company (Qingdao, China). Acetonitrile and Acetic acid were purchased from the Chengdu Kelong Chemical Co., Ltd. (Chengdu, China). Chitosan (Deacetylation Degree ≥ 90%) was obtained from Shandong Haiyihua Biotechnology Co., Ltd. (Jinan, China). Fe_3_O_4_ was provided by Shanghai Yien Chemical Technology Co., Ltd. (Shanghai, China). Glutaraldehyde was supplied from Shanghai Zhanyun Chemical Co., Ltd. (Shanghai, China). Methanol was obtained from Sinopharm Chemical Reagent Co., Ltd. (Beijing, China). Phenolphthalein was acquired from Shanghai Zhongqin Chemical Reagent Co., Ltd. (Shanghai, China). Folin-phenol reagent was purchased from Tianjin Guangfu Technological Development Co., Ltd. (Tianjin, China). White wine (alcohol: 12.5%, grape variety: chardonnay, type: dry, 750 mL) and red wine (alcohol: 12.5%, grape variety: cabernet, type: dry, 750 mL) both were gained from COFCO Great Wall Wine Co., Ltd. (Beijing, China).

### 2.2. Preparation of Magnetic Nano-Fe_3_O_4_@CTS Adsorbent

The novel magnetic adsorbent was prepared by the following procedure. A total of 6 g chitosan (CTS) and 6 g polyvinyl alcohol (PVA) were dissolved in 150 mL 2% glacial acetic acid solution and 150 mL distilled water, respectively. After that, 6 g Fe_3_O_4_ nanoparticles were dispersed in the mixed solution while stirring magnetically for 3 h at 25 ± 1 °C until the mixture became a uniform gel solution. The gel solution was slowly dropped into the sodium hydroxide solution to form a homogeneous gel ball, and then soaked for 1 h. The gel balls were rinsed and stirred in the 0.012 mmol glutaraldehyde solution for 12 h (25 ± 1 °C). The final products were cleaned with distilled water repeatedly and dried for 24 h at 70 °C. The chitosan was labeled with magnetic Fe_3_O_4_ nanoparticle modification as nano-Fe_3_O_4_@CTS, and the fabrication steps and synthesis strategy were illustrated in Figure 1.

### 2.3. Characterization of Nano-Fe_3_O_4_@CTS Adsorbents

The functional groups and the synthesis of nano adsorbents were confirmed using Fourier Transform Infrared Spectroscopy (FTIR-650, Tianjin Guangdong Sci.&Tech. Co., Ltd., Tianjin, China), and the range of analysis was 4000−500 cm^−1^. The magnetic properties of Fe_3_O_4_ and nano-Fe_3_O_4_@CTS were marked using Vibrating Sample Magne (VSM, 7404/8604, LakeShore, Columbus, OH, USA), and the magnetic field ranged from −22,500–22,500 Oe. The crystal structure of the synthesized nano adsorbents was investigated by the X-ray Diffraction (XRD, MSAL XD-3, Puxi General Instrument Co., Ltd., Beijing, China), using CuKα radiation (λ = 0.15418 nm), and the scanning range was 10–80°. The particle size of nano-Fe_3_O_4_@CTS was evaluated in distilled water by a Mastersizer Hydro 2000SM (Malvern Panalytical Ltd., Shanghai, China) laser diffraction particle size analyzer.

### 2.4. Screening of Optimization Conditions

To obtain high removal efficiency, the adsorption conditions were optimized. Briefly, 1.0, 1.5, 2.0, 2.5, 3.0 g of nano-Fe_3_O_4_@CTS were dissolved in 20 mL OTA solution (C_OTA_ = 0.5 μg/mL, pH = 3, 4, 5, 6, 7), respectively. Then, the mixed solution was adsorbed in a shaker for 20, 30, 40, 50, 60 min, and the temperature was controlled at 25, 30, 35, 40, and 45 °C, respectively. The nano-Fe_3_O_4_@CTS nanoparticles were then separated from the solution by using an external magnet, and the supernatant was collected. The supernatant was filtered through a 0.22 μm nylon membrane and OTA concentrations were quantified by high-performance liquid chromatography fluorescence detection (HPLC-FLD). Equations (1) and (2) were used to determine the adsorbents’ *q_e_* (adsorption capacity) and *Q* (removal rate):(1)$$q_{e}=\frac{\left(c_{0}-c_{e}\right)\times V}{m}$$
(2)$$Q=\frac{\left(c_{0}-c_{e}\right)}{c_{0}}\;\times \;100\%$$

In the equation, *c*_0_ and *c_e_* represent the initial and final concentrations of OTA (g/mL), *V* represents the volume (mL), and *m* represents the doses (g).

### 2.5. Adsorption Kinetics

Three models (the Weber–Morris model, the pseudo-first order model, and the pseudo-second order model) were used to examine the adsorption mechanism. A total of 1 g nano-Fe_3_O_4_@CTS nanoparticles were added in 20 mL OTA standard solution; then, the mixed solution was oscillated at 120 rpm for 10, 20, 30, 40, 50, 60, 70, 80 min, respectively, when the temperature was 25 °C. The nano-Fe_3_O_4_@CTS nanoparticles were recovered by an adding magnet, and the supernatant was taken to determine the concentration of OTA by HPLC-FLD. The three kinetic equations are given as follows:

(1) Pseudo-first order equation
(3)$$\ln\left(q_{e}-q_{t}\right)=\ln q_{e}-k_{1}t$$

(2) Pseudo-second order equation(4)$$\frac{t}{q_{t}}=\frac{t}{q_{e}}+\frac{1}{k_{2}q_{2}^{2}}$$

(3) Weber–Morris equation
(5)$$q_{t}=k_{3}\;\times \;t^{0.5}+c$$

In the equation, *q_e_* and *q_t_* represent, respectively, the adsorption capacity at equilibrium and at a specific time (μg/g). The rate constants *k*_1_ and *k*_2_ correspond to the pseudo-first order and pseudo-second order reactions. *k*_3_ denotes the rate constant for intra-particle diffusion (μg/g/min^0.5^).

### 2.6. Adsorption Isotherms

After adding 1 g nano-Fe_3_O_4_@CTS nanoparticle in 20 mL concentrations of 0.5, 1.0, 1.5, 2.0, and 2.5 μg/mL OTA working solution, and the mixture was adsorbed for 30 min at 25 °C, 35 °C, and 45 °C. After separation, the supernatant was taken to determine the concentration of OTA, and the equilibrium adsorption capacity (*q_e_*) was calculated. The Langmuir, Freundlich, and Temkin isotherm equations were applied to analyze the experimental data.

(1) Langmuir equation(6)$$\frac{c_{e}}{q_{e}}=\frac{c_{e}}{q_{m}}+\frac{1}{K_{L}\times q_{m}}$$

(2) Freundlich equation
(7)$$lnq_{e}=lnK_{f}+\frac{lnc_{e}}{n}$$

(3) Temkin equation
(8)$$q_{e}=\frac{RT}{b}\left(K_{T}lnc_{e}\right)$$

In the equation, *c_e_* represents the equilibrium concentration of OTA (μg/mL), *q_e_*, *q_m_* represent, respectively, the equilibrium adsorption capacity and maximum adsorption capacity (μg/g). *K_L_*, *K_f_* and *n* represent, respectively, the Langmuir, the Freundlich constants. *RT/b* is associated with the adsorption heat (μg/g), *R* = 8.314 J/mol·K.

### 2.7. Thermodynamic Parameters

The spontaneity of the adsorption process can be determined by the Gibbs free energy change (*ΔG°*, KJ/mol), enthalpy change (*ΔH°*, KJ/mol), and entropy change (*ΔS°*, KJ/(mol·K)). The *ΔG°*, *ΔH°* and *ΔS°* are calculated through the following Equations (9)–(11).(9)$$lnc_{e}=\frac{\Delta H{}^{\circ}}{RT}+K$$(10)ΔG°=−nRT(11)$$\Delta S{}^{\circ}=\frac{\Delta H{}^{\circ}-\Delta G{}^{\circ}}{T}$$
where *Ce* is the concentration of OTA at equilibrium (ng/mL), *R* = 8.314 J/mol∙K, *K* is the distribution coefficient, *T* represents the temperature (*K*), and *ΔG°*, *ΔS°*, and *ΔH°* are the Gibbs free energy change, entropy change, and enthalpy change in adsorption reaction, respectively.

### 2.8. HPLC-FLD Analysis

High-performance liquid chromatography with a fluorescence detector (Agilent 1260 Infinity II, Santa Clara, CA, USA) was used to quantify the OTA content. The analysis of OTA was finished by a C18 column (4.6 mm × 250 mm × 5 μm) when the column temperature was 35 °C. The mobile phase consists of water and acetonitrile (49:51, *v*/*v*). The injection volume was 20 µL and the flow rate was 1 mL/min. The excitation and emission wavelengths of the fluorescence detector are 333 and 460 nm, respectively.

### 2.9. Removal of OTA from Wine

The following protocol was used to determine how to extract and clean up OTA in red and white wine. OTA standard solution was added to white and red wine to obtain spiked solutions with final concentrations of 10, 20, and 30 ng/mL. A total of 1 g of nano-Fe_3_O_4_@CTS nanoparticles were added to 20 mL of spiked wine solution, respectively, and shaken for 60 min at 25 °C, 120 rpm. After OTA removal, the nano-Fe_3_O_4_@CTS nanoparticles were separated by an external magnetic field, and 1 mL of supernatant was taken in a 25 mL volumetric flask and diluted with eluent (15% sodium chloride-2% sodium bicarbonate solution). The extraction process was assisted by ultrasound for 5 min and centrifuged for 10 min (5000 r/min). Then, the 10 mL supernatant was purified with an Ochratoxin A immunochromatographic affinity column (IAC, Pribolab, Qingdao, China) and washed with 10 mL PBS and ultrapure water in turn. The 2 mL eluent (methanol/acetic acid = 98/2, *v*/*v*) was used to release the OTA from IAC, and the eluent was vortexed for 30 s and filtered through a 0.22 μm filter membrane. OTA concentrations in the eluent were quantified by HPLC-FLD.

### 2.10. Wine Analysis

#### 2.10.1. Measurement of Color Value

The change in absorbance value of wine samples was determined by UV spectrophotometer, and distilled water was used as a blank control. Each absorbance was measured at least 3 times. The color value was concluded from the absorbance at 420 nm. The average of the three values was defined as the color value [21].

#### 2.10.2. Measurement of Transmittance

Transmittance was expressed using the absorbance. With distilled water as a control, the transmittance was determined based on the absorbance at 625 nm and was measured at least three times for each sample [22].

#### 2.10.3. Measurement of Total Soluble Solids

TSS content of wine was measured using a hand-held Abbe refractometer. A total of 1 mL of wine mixture was dropped onto a refractometer, and three readings were taken for the wine mixture.

#### 2.10.4. Measurement of Total Polyphenol

Total polyphenol was measured by Folin–Ciocalteu colorimetry using the method from Xin et al. with modification [23]. A total of 0.5 mL of wine mixture, 5 mL of distilled water, and 1.5 mL of Folin–Ciocalteu reagent were added to a 25 mL colorimetric tube, and reacted for 5 min. Then, 6 mL 20% sodium carbonate solution was added, and the mixture was reacted unceasingly for 60 min. The absorbance at 760 nm was measured with distilled water as a blank control. Thus, the total phenolic content was expressed as g gallic acid (GA) kg^−1^.

#### 2.10.5. Measurement of Total Flavonoid Content

The total flavonoid content of the wine was ascertained following the methodology of Jia et al. with minor modification [24]. A total of 2 mL wine mixture was homogenized in 10 mL of ethanol (95%), and it was centrifuged at 3000 rpm for 5 min. After that, 2 mL of 5% sodium nitrite solution and 10% aluminum nitrate solution, and 8 mL of 1 mol/L sodium hydroxide solution were added to the mixed solution in turn and allowed to stand for 15 min. The absorbance of the solution at 510 nm was measured. Therefore, the total flavonoid was calculated and expressed as rutin content using the calibration curve created from 0.5 to 5 μg/mL.

#### 2.10.6. Measurement of Total Acidity

The total acid content of the wine sample was estimated using the Han et al. technique [25]. Two drops of 1% phenolphthalein were added to 20 mL of wine mixture and titrated with a sodium hydroxide solution (1.0 mol/L). When the color of the solution became pink and did not fade within 30 s, the volume of sodium hydroxide required was recorded. Titration was repeated 3 times for each sample, and the final value was the average of three. The total acidity was expressed as tartaric acid (g/L).

### 2.11. Statistical Analysis

HPLC-FLD was used for quantitative analysis of OTA concentration. All analyses were carried out in triplicate and the results were presented as a mean value ± standard errors, and statistical analysis was performed with Excel 2021. The standard errors are presented in the form of error bars. Origin 2018 was used to draw line charts in articles.

## 3. Results and Discussion

### 3.1. Characterization of Nano-Fe_3_O_4_@CTS

Numerous characterization ways including XRD (Figure 2A), VSM (Figure 2B) and FT-IR (Figure 2C) were employed to characterize the behavior of nano-Fe_3_O_4_@CTS. The pure Fe_3_O_4_ nanoparticles produced six distinguishing peaks (2θ = 30.02°, 35.18°, 42.84°, 53.28°, 56.78°, and 62.40°) [26]. The characteristic diffraction peaks of nano-Fe_3_O_4_@CTS was akin to the XRD pattern of Fe_3_O_4_, which proved nano-Fe_3_O_4_@CTS has a similar character and structure to pure Fe_3_O_4_.

In FT-IR spectra of Fe_3_O_4_, Fe-O vibration appeared at 580 cm^−1^ [12]. For the chitosan, the region at 3500–3200 cm^−1^ was linked with the O-H and N-H stretching, 1627 cm^−1^ (bending), 1394 cm^−1^ (C-O-C), and 1076 cm^−1^ (C-N stretching) [27]. The FT-IR spectra of nano-Fe_3_O_4_@CTS appeared at peaks of CTS and Fe_3_O_4_, proving the presence of CTS and Fe_3_O_4_. In addition, the particle size distribution of nano-Fe_3_O_4_@CTS conforms to the Gaussian distribution (R^2^ = 0.9340), and the average particle size is recorded as 162.81 nm (Figure 2D). FT-IR and XRD results reveal that CTS and Fe_3_O_4_ nanospheres successfully combined. Magnetism plays a crucial role in the separation process [5]. The magnetic properties of pure Fe_3_O_4_ and nano-Fe_3_O_4_@CTS were studied by VSM, and the hysteresis loops are given in Figure 2B. It can be inferred from the S-shaped curve depicted in Figure 2B that Fe_3_O_4_ and nano-Fe_3_O_4_@CTS nanoparticles both have superparamagnetic properties [28]. The saturation magnetization of nano-Fe_3_O_4_@CTS was calculated to be approximately 26.406 emu/g, which was lower than pure Fe_3_O_4_ nanoparticles (Ms = 79.361 emu/g). Despite having a lower saturation magnetization than pure Fe_3_O_4_, nano-Fe_3_O_4_@CTS could be rapidly isolated from the mixed solution using the external magnetic field, to realize rapid solid–liquid separation and greatly improve separation efficiency.

### 3.2. Adsorption Kinetics

Adsorption kinetics are the main methods for examining adsorption effectiveness [11]. The pseudo-first order kinetic is applied to simple physical adsorption, and the number of vacancies is positively correlated with the occupancy rate of the adsorption region [29]. The pseudo-second order kinetic reflects processes such as liquid film diffusion, surface adsorption, and intraparticle diffusion, and is a combination of physical adsorption and chemical adsorption [30].

The optimization of adsorption conditions was conducted, and the experimental results are presented in Figure 3. At the optimal adsorption time of 30 min, adsorption temperature of 25 °C, adsorbent dosage of 1 g, and solution pH of 5, the kinetic studies were investigated between OTA and nano-Fe_3_O_4_@CTS to analyze the transport mechanism [31]. The dynamic parameters of the three models are given in Table 1. The pseudo-second order kinetic model’s correlation coefficient value (R^2^ = 0.9991) is higher than that of Weber–Morris model (0.3305) and pseudo-first order kinetic model (0.5920), suggesting the adsorption of OTA by nano-Fe_3_O_4_@CTS is more suitable to be described by the pseudo-second order model.

According to previous research on the physical adsorption process, it is concluded that the adsorption steps of OTA on nano-Fe_3_O_4_@CTS were divided into three main stages [13]. At first, OTA was adsorbed on the outer surface of the nano-Fe_3_O_4_@CTS; then OTA diffused in the nano-Fe_3_O_4_@CTS particles; finally, OTA was adsorbed on the inner surface of the nano-Fe_3_O_4_@CTS [32]. Since the number of adsorption sites per unit mass of nano-Fe_3_O_4_@CTS surface remains essentially constant, the adsorption sites of nano-Fe_3_O_4_@CTS could be fully combined with OTA [13]. As the adsorption proceeded, the available adsorption sites were reduced accordingly, so physical adsorption was no longer in charge of the adsorption process. According to Sharma et al. [31], the fitted curve for the intraparticle diffusion model does not cross the coordinate origin, demonstrating that intraparticle diffusion is not the main step in the adsorption process. In the report by Annadurai et al. [33], there is an ionic interaction between the sulfonic acid group (-SO_3_) of the dye and the amino group of chitosan. Therefore, as adsorption progresses, we speculate physical adsorption is no longer the main step, and the weak chemisorption between -NH_2_ of the chitosan and -COOH of the OTA takes place by ionic interaction [28]. Drawing from the aforementioned findings, it can be concluded that the adsorption mechanism of OTA involves physical adsorption and chemisorption [34].

### 3.3. Adsorption Isotherm

The Langmuir isotherm model states that adsorption happens on homogeneous surfaces and there are no lateral interactions [35]. The Freundlich isotherm is a multilayer adsorption model and addresses the adsorption regions of heterogeneous surfaces or surfaces with different affinities [36]. The Temkin isotherm postulates that the adsorption heat falls linearly with the adsorbent’s surface coverage [33]. To understand the feature of how OTA are adsorbed onto the nano-Fe_3_O_4_@CTS at the equilibrium, the three linear equation models (Langmuir, Freundlich, and Temkin isotherms) were utilized to evaluate the isotherm parameters. Table 2 shows the correlation coefficients at various temperatures. The Langmuir model fits the date better than the other two models because the correlation coefficient R^2^ (0.9754, 0.9575, and 0.9970) is generally higher than that of the Freundlich model (0.9735, 0.9591, 0.9552) and Temkin model (0.8415, 0.8681, 0.7382). Therefore, the adsorption of OTA by nano-Fe_3_O_4_@CTS follows the Langmuir model, suggesting that OTA adsorption on nano-Fe_3_O_4_@CTS was the monolayer adsorption. The maximum adsorption capacities *q_m_* were 5018.07, 4484.3, and 3988.83 ng/g at 25 °C, 35 °C, and 45 °C, respectively. The *q_m_* values decrease with increasing temperature, showing that OTA adsorption onto nano-Fe_3_O_4_@CTS is more favorable at lower temperatures.

In the Freundlich model, the *n* value represents the type of isotherm. When 0 < 1/n < 1, adsorption is helpful; when 1/n = 1, adsorption is irreversible; and when 1/n > 1, adsorption is unfavorable [37]. The 1/n value is found to be 3.8081, 3.8865, and 4.3141, which means the adsorption of OTA onto nano-Fe_3_O_4_@CTS nanoparticles is not conducive to multilayer adsorption. The *K_T_* reflects the maximum binding energy between molecules, and a larger *K_T_* value demonstrated the more powerful interaction between the OTA and nano-Fe_3_O_4_@CTS [38]. The *K_T_* values in the temperature range of 25–45 °C were 5.2049, 4.6297, and 4.6455, respectively, which decreased with increasing temperature. All this indicates that the interaction between OTA and nano-Fe_3_O_4_@CTS becomes weaker as the temperature increases, which further explains that high temperature is not conducive to adsorption.

### 3.4. Thermodynamic Parameters

The thermodynamic parameters (*ΔG°*, *ΔH°*, *ΔS°*) were researched and calculated to learn more about the energy changes [39]. Table 3 gives the computed values of *ΔG°*, *ΔH°* and *ΔS°* when the initial concentration of OTA standard solution is 0.5, 1, 1.5, 2, and 2.5 µg/mL and with temperatures of 25, 35, and 45 °C. Usually, when the *ΔG°* value falls between −20 and 0 kJ/mol, the process is designated as physical adsorption. A *ΔG°* value between −400 and −80 kJ/mol indicates that the process is chemisorption [40]. The *ΔG°* value ranged from −0.660 to −0.613 kJ/mol when the temperature rose from 25 °C to 45 °C, indicating the adsorption process was spontaneous and favorable at low temperatures. When *ΔH°* values were negative, it indicated the adsorption process was the exothermic process, and the adsorption of the OTA onto the nano-Fe_3_O_4_@CTS is due to the exothermic nature of the adsorption process [41]. The adsorption and desorption occur simultaneously when OTA is adsorbed on the nano-Fe_3_O_4_@CTS nanoparticles, which leads to the negative *ΔH°* [42]. *ΔS°* was negative as well and illustrated the reduced randomness at the nano-Fe_3_O_4_@CTS-OTA solution interface during the adsorption process [43].

### 3.5. Extraction and Removal of OTA from Samples

To evaluate the practical applicability of the nano-Fe_3_O_4_@CTS nanoparticle, the content of OTA in wine was analyzed before and after OTA removal. OTA was added to the sample of wine to obtain a final solution concentration of 10 ng/mL, 20 ng/mL, and 30 ng/mL. As depicted in Table 4, after adsorption treatment, the removal rate in red and white wine rose with increasing solution concentration. For red wine, the removal rate was 68.4–79.4%. However, for white wine, a lower removal efficiency was observed (46.0–63.3%). Activated carbons were used to explore its removal effect in white and red wine, the results revealed a higher removal rate for the white wine, which is contrary to our results. One of the reasons may be that the structural characteristics of ACs are crucial in the complicated matrix [12]. Olivares et al. [44] Found that ACs have a higher removal rate in red wine when ACs have larger size pores (54%). Therefore, the different manifestations found in the literature may be attributed to nano-Fe_3_O_4_@CTS having different physical and chemical properties from ACs.

### 3.6. Wine Analysis

To investigate how applying nano-Fe_3_O_4_@CTS nanoparticles affects the nutritional quality of wine, the composition of white and red wines was measured and shown in Table 5. After the removal of OTA by nano-Fe_3_O_4_@CTS nanoparticles, the color value was significantly decreased by 61.82% in red wine, while it was increased by 82.84% in white wine. The decline in color value may be related to the decrease in total phenolic content [12]. It was vital to determine the concentrations of total phenols and flavonoids in wine, and to better study its possible antioxidant properties [25]. The use of nano-Fe_3_O_4_@CTS on wines resulted in a decrease in total phenol from 11.118 µg/mL to 5.807 µg/mL (red wine), and from 3.350 µg/mL to 0.975 µg/mL (white wine). The functional groups (-OH and -NH_2_) in the chitosan can interact with hydroxyl (-OH) and carboxyl (-COOH) groups in phenolic substances through hydrogen and chemical bonds [45]. Some studies have shown that flavonoids have a visible impact on the bitterness of wine and have antioxidant properties [46]. The flavonoid content dropped by 40.92% (red wine) and 54.43% (white wine), respectively. The soluble content in red wine and white wine decreased by 7.95% and 7.46%, respectively. Transmittance is one of the important indicators to characterize the clarification effect of wine. The higher the transmittance, the better the clarification effect [47]. After adsorption treatment, the transmittance in red wine decreased, while the transmittance in white wine increased. The total acid content of red wine and white wine decreased because the adsorbent has affinity and interaction with some acids in the wine, or it may combine with some organic acids to form precipitates, which led to the drop in the total acid content of the wine [48].

### 3.7. Removal Mechanism of OTA by Nano-Fe_3_O_4_@CTS

Based on the experimental findings, Figure 4 summarizes and illustrates the OTA adsorption mechanism on the nano-Fe_3_O_4_@CTS nanoparticle. The surface of Fe_3_O_4_ is uniformly modified with chitosan, which solves the problem that chitosan is hard to isolate from the solution. Combining the isotherm data, the R^2^ value of the Langmuir model is typically higher than those of the Temkin and Freundlich models. The negative *ΔG°*, *ΔH°*, and *ΔS°* values suggest that the adsorption process was spontaneous, exothermic, and reduced in degree of freedom. Otherwise, the *ΔG°* value falls between −20 and 0 kJ/mol, indicating the adsorption process is designated as physical adsorption, and the porosity structure of the nano-Fe_3_O_4_@CTS adsorbent had an active effect on physical adsorption [36]. As is well known, physical adsorption requires a lower activation energy, resulting in faster adsorption and desorption rates, and adsorption can reach equilibrium within a short time, while chemical adsorption requires a longer time since chemical bonds need to form. Liang et al. showed that the adsorption of aflatoxin B1, zearalenone, and T-2 toxin by montmorillonite and attapulgite quickly reached equilibrium in less than 1 h [49]. In this study, the removal rate of OTA by nano-Fe_3_O_4_@CTS can reach a high level at 30 min (Figure 3D), indicating that OTA can be quickly adsorbed to the surface of nano-Fe_3_O_4_@CTS. As the adsorption proceeded, the available adsorption sites became less accordingly, resulting in a reduction in the removal rate. The kinetic parameters suggest that the adsorption process follows the pseudo-second order model. Therefore, the removal of OTA on the nano-Fe_3_O_4_@CTS nanoparticles is not only dominated by physical adsorption, but also the weak chemical adsorption between -NH_2_ (CTS) and -COOH (OTA) takes place by ionic interaction. In summary, the mechanism of OTA adsorption by nano-Fe_3_O_4_@CTS nanoparticles was a combination of physisorption and chemisorption, while within 30 min the adsorption was dominated by spontaneous exothermic and nonspecific physisorption after that chemisorption came into play.

## 4. Conclusions

In this study, nano-Fe_3_O_4_@CTS nanoparticles were developed and applied for OTA removal in wine. The thermodynamic data suggested that the adsorption process follows the Langmuir isotherm and monolayer adsorption is the primary mode of adsorption. The process was dominated by a spontaneous exothermic physical adsorption process based on the negative *ΔG°*, *ΔS°*, and *ΔH°*. In addition, the adsorption of OTA by nano-Fe_3_O_4_@CTS nanoparticles followed a second-order kinetic model. It can be concluded before 30 min that adsorption was mainly physisorption dominated by porosity. As the adsorption proceeds, the carboxyl group(-COOH) on OTA reacts with the amino group(-NH_2_) on chitosan. Based on the isotherm and kinetic data, it is possible to infer that the adsorption mechanism of OTA on nano-Fe_3_O_4_@CTS is initially based on physisorption and followed by chemisorption. The nano-Fe_3_O_4_@CTS achieved a good OTA removal rate (red wine: 68.4–79.4%, white wine: 46.0–63.3%) from wine, which can be easily and quickly separated from red wine. The nano-Fe_3_O_4_@CTS can effectively reduce the concentration of OTA, but OTA cannot be completely removed. The final OTA concentration in wines may fluctuate by fermentation process and treatments before bottling (clarification, filtration, etc.). In future studies, nano-Fe_3_O_4_@CTS can be specifically modified to reduce its negative impact on wine and improve application range and removal efficiency.

## Figures and Tables

**Figure 1 foods-14-00666-f001:**
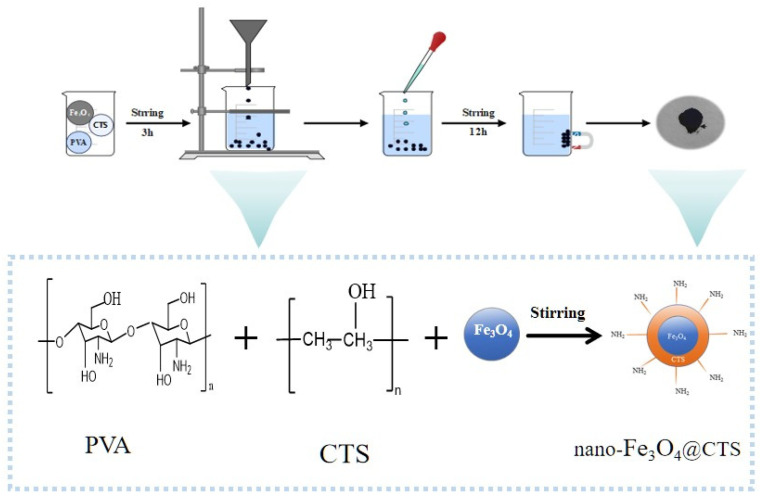
Schematic illustrations of the synthesis strategy used to prepare nano-Fe_3_O_4_@CTS.

**Figure 2 foods-14-00666-f002:**
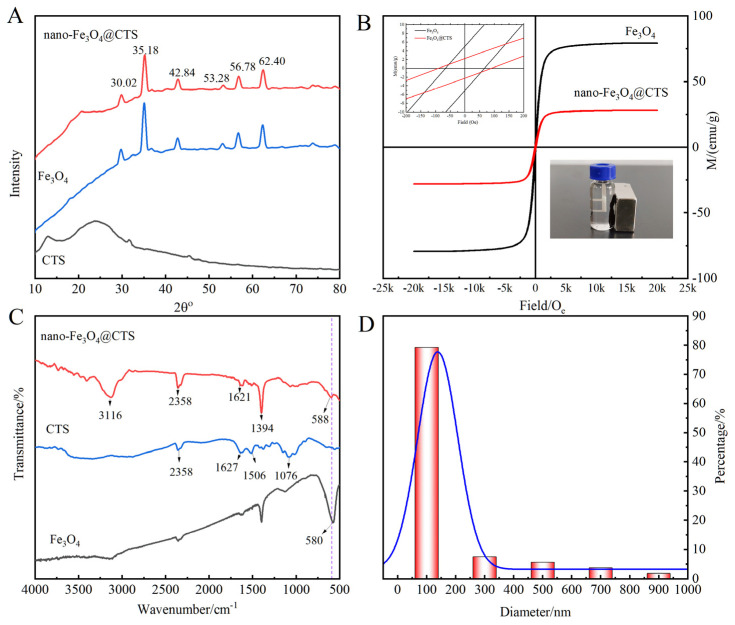
(**A**) XRD of CTS, Fe_3_O_4_ and nano-Fe_3_O_4_@CTS; (**B**) VSM of Fe_3_O_4_ and nano-Fe_3_O_4_@CTS; (**C**) FT-IR of CTS, Fe_3_O_4_ and nano-Fe_3_O_4_@CTS; (**D**) Particle size distribution of nano-Fe_3_O_4_@CTS.

**Figure 3 foods-14-00666-f003:**
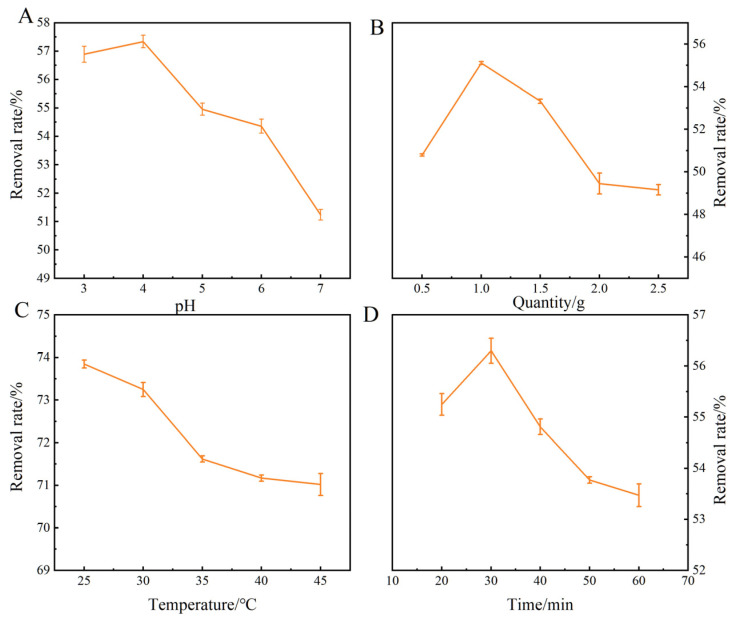
(**A**) Effect of pH, (**B**) Effects of adsorbent dosage, (**C**) Effect of adsorption temperature, (**D**) Effect of adsorption time. The error bar is expressed by standard deviation.

**Figure 4 foods-14-00666-f004:**
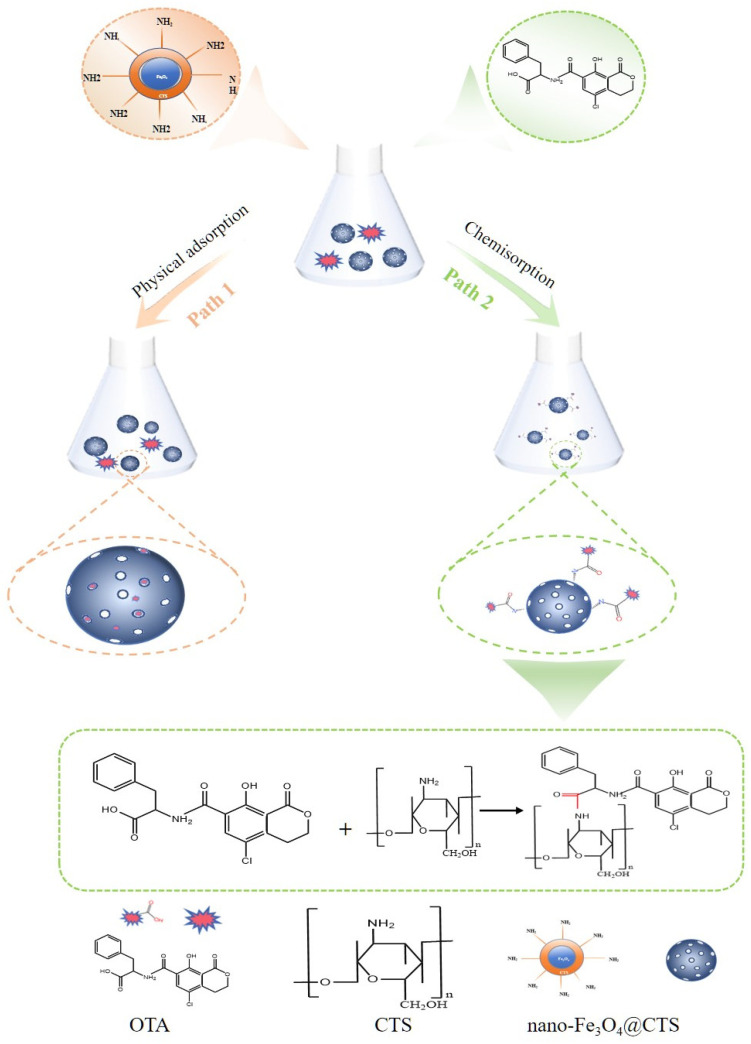
Removal schematic of OTA with nano-Fe_3_O_4_@CTS.

**Table 1 foods-14-00666-t001:** Kinetic parameters for adsorption of OTA on nano-Fe_3_O_4_@CTS.

Kinetic Models	Fitting Equation	Kinetic Parameters		
Pseudo-first order equation	Ln (177.86 − q_t_) = 5.181 − 3.57 × 10^−2^t	R_1_^2^ 0.5920	K_1_(1/min) 3.57 × 10^−2^	q_e_ (ng/g) 177.86
Pseudo-second order equation	t/q_t_ = t/2259.16 + 3.929 × 10^−4^	R_2_^2^ 0.9991	k_2_ (g/ng/min) 4.832 × 10^−4^	q_e_ (ng/g) 2259.16
Weber-Morris equation	q_t_ = 27.77t^0.5^ + 2035.17	R_3_^2^ 0.3305	k_3_(ng/g/min^0.5^) 27.77	C (ng/g) 2035.17

**Table 2 foods-14-00666-t002:** Model fitting and isotherms constants of adsorption isotherm model.

Model	Temperature/°C	Fitting Equation	Parameters		
			K_L_	q_m_	R^2^
Langmuir model	25	$\frac{1}{q_{e}}=0.0714\frac{1}{c_{e}}+0.1993$	2.7910	5018.07	0.9754
	35	$\frac{1}{q_{e}}=0.0881\frac{1}{c_{e}}+0.2230$	2.5312	4484.30	0.9575
	45	$\frac{1}{q_{e}}=0.0944\frac{1}{c_{e}}+0.2507$	2.6557	3988.83	0.9970
			K_f_	n	R^2^
Freundlich model	25	lnq_e_ = 3.8077lnc_e_ + 7.9156	2.7397	0.2626	0.9735
	35	lnq_e_ = 3.8858lnc_e_ + 7.5510	1.9025	0.2573	0.9591
	45	lnq_e_ = 4.3133lnc_e_ + 8.1607	3.5007	0.2318	0.9552
			K_T_	β	R^2^
Temkin model	25	q_e_ = 68.4555lnc_e_ + 112.9230	5.2049	36.2107	0.8415
	35	q_e_ = 67.9942lnc_e_ + 104.1987	4.6297	37.6791	0.8681
	45	q_e_ = 72.5497lnc_e_ + 111.4258	4.6455	36.4591	0.7382

**Table 3 foods-14-00666-t003:** Thermodynamic parameters of adsorption of OTA by nano-Fe_3_O_4_@CTS.

Concentration (µg/mL)	ΔH° (KJ/mol)	ΔG° (KJ/mol)			ΔS° (KJ/(mol*K))		
		25 °C	35 °C	45 °C	25 °C	35 °C	45 °C
0.5	−4.195	−0.651	−0.660	−0.613	−0.0118	−0.0115	−0.0113
1.0	−3.981				−0.0112	−0.0107	−0.0106
1.5	−4.391				−0.0125	−0.0121	−0.0119
2.0	−3.419				−0.0095	−0.0090	−0.0088
2.5	−1.483				−0.0028	−0.0027	−0.0027

**Table 4 foods-14-00666-t004:** Removal of OTA with nano-Fe_3_O_4_@CTS in red and white wine.

Sample	Initial Concentration (ng/mL)	Final Concentration (ng/mL)	Removal Rate (%)
	10	3.2	68.4
Red wine	20	5.2	74.2
	30	6.2	79.4
	10	5.4	46.0
White wine	20	7.8	61.1
	30	11.0	63.3

**Table 5 foods-14-00666-t005:** Composition of wine before and after removal of OTA by nano-Fe_3_O_4_@CTS.

		Color Vale	Total Phenol	TSS	Transmittance	Total Flavonoid	Total Acid
Red wine	Before	2.606 ± 0.0014	11.118 ± 0.0182	7.803 ± 0.0205	0.938 ± 0.0043	52.741 ± 0.5014	5.766 ± 0.5340
	After	0.995 ± 0.0017	5.807 ± 0.0232	7.177 ± 0.0205	0.467 ± 0.0012	31.162 ± 0.8684	0.546 ± 0.1370
White wine	Before	0.134 ± 0.0012	3.350 ± 0.0168	6.697 ± 0.0047	0.007 ± 0.0005	7.665 ± 0.3762	4.442 ± 0.0430
	Before	0.245 ± 0.0026	0.975 ± 0.0019	6.683 ± 0.0624	0.143 ± 0.0026	3.493 ± 0.4978	1.352 ± 0.1673

## Data Availability

The original contributions presented in this study are included in the article. Further inquiries can be directed to the corresponding author.

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
