# Peer review of "Removal Capacity and Mechanism of Modified Chitosan for Ochratoxin A Based on Rapid Magnetic Separation Technology"

_foods, 2025, doi:10.3390/foods14040666_

Round 1
Reviewer 1 Report
Comments and Suggestions for Authors
This study develops nano-Fe3O4@CTS nanoparticles for ochratoxin A (OTA) removal from wine, achieving high efficiency through a combination of physical and chemical adsorption. Adsorption follows the Langmuir isotherm and pseudo-second-order kinetics, favoring lower temperatures. While effective, nano-Fe3O4@CTS impacts wine's nutritional quality. Future improvements aim to enhance specificity, minimize quality loss, and explore broader food applications. To strengthen the article authors should consider the below comments while revising.
1. What is the specific contribution of physisorption and chemisorption to the total adsorption capacity of nano-Fe3O4@CTS?
2. How does the adsorption efficiency vary with different initial OTA concentrations and in the presence of other wine components (e.g., polyphenols, tannins, and organic acids)?
3. Can nano-Fe3O4@CTS effectively adsorb OTA in food matrices beyond wine, such as beer, juices, or grains, and how do matrix complexities affect adsorption?
4. What is the long-term stability, reusability, and cost-effectiveness of nano-Fe3O4@CTS, particularly under repetitive adsorption-desorption cycles?
5. How do the size distribution, morphology, and synthesis methods of nano-Fe3O4@CTS influence adsorption performance, production cost, and scalability?
6. What are the optimal conditions (e.g., pH, temperature, adsorbent dosage) for OTA removal across different wine types and food matrices?
7. How does the adsorption process compare with existing mycotoxin removal methods (e.g., activated carbon) regarding efficiency, cost, safety, and environmental impact?
8. What are the regulatory and consumer safety implications of using nano-Fe3O4@CTS in food and beverage applications, particularly regarding residues, health effects, and compliance with food safety standards?
9. Are there potential competitive or synergistic interactions between OTA and other mycotoxins, contaminants, or wine components that might influence adsorption behavior?
10. Can nano-Fe3O4@CTS be further modified or combined with other materials to improve specificity, adsorption capacity, and versatility for removing OTA and other mycotoxins?
Author Response
This study develops nano-Fe3O4@CTS nanoparticles for ochratoxin A (OTA) removal from wine, achieving high efficiency through a combination of physical and chemical adsorption. Adsorption follows the Langmuir isotherm and pseudo-second-order kinetics, favoring lower temperatures. While effective, nano-Fe3O4@CTS impacts wine's nutritional quality. Future improvements aim to enhance specificity, minimize quality loss, and explore broader food applications. To strengthen the article authors should consider the below comments while revising.
- What is the specific contribution of physisorption and chemisorption to the total adsorption capacity of nano-Fe3O4@CTS?
Response: Thank you very much for your advice. We explored the adsorption type of the adsorbent through thermodynamic and kinetic experiments. Physical adsorption requires a lower activation energy, spontaneous, and adsorption can reach equilibrium within a short time. While chemical adsorption requires a longer time since chemical bonds need to form. Thermodynamic data showed that the correlation coefficient R2 (0.9754, 0.9575, and 0.9970) of the Langmuir model is higher than that of the Temkin (0.9735, 0.9591, 0.9552) and Freundlich models (0.8415, 0.8681, 0.7382) When the temperature is 25 °C, 35 °C, 45 °C, respectively. The values of ΔG°, ΔH°, and ΔS° were all negative, indicating that the adsorption of OTA by the nano-Fe3O4@CTS was spontaneous and exothermic. In addition, the ΔG° value is between -20 and 0 k J / mol, indicating that the process is physical adsorption. According to the single factor experiment, we speculated that the pore structure of the nano-Fe3O4@CTS had a positive effect on physical adsorption within 30 min. With the extension of adsorption time, the available sites on the surface of the nano-Fe3O4@CTS gradually decreased, and ion interaction occurred between the amino group (CTS) and the carboxyl group (OTA), showing chemical adsorption. In short, the adsorption of OTA by the nano-Fe3O4@CTS is mainly dominated by physical adsorption. When the adsorption time is prolonged, chemical adsorption will occur.
- How does the adsorption efficiency vary with different initial OTA concentrations and in the presence of other wine components (e.g., polyphenols, tannins, and organic acids)?
Response: Polyphenols, tannins, and organic acids etc. are important nutrients in wine, which may interfere with the adsorption effect of nano-Fe3O4@CTS.
In order to evaluate the application of the nano-Fe3O4@CTS in the actual wine sample, we analyzed the content of OTA before and after adsorption, and calculated the removal rate. Since OTA was not detected in the wine stock solution, different volumes of OTA standard solution were added to wine to obtain a final solution concentration of 10 ng/mL, 20 ng/mL and 30 ng/mL. When the initial concentration of OTA increased, the removal rate in white and red wine increased (red wine: 68.4~79.4%, white wine: 46.0~63.3%). However, due to the influence of wine nutrients, the removal rate of OTA in wine samples is relatively lower than that of OTA standard solution.
- Can nano-Fe3O4@CTS effectively adsorb OTA in food matrices beyond wine, such as beer, juices, or grains, and how do matrix complexities affect adsorption?
Response: According to the research about dietary exposure to food mycotoxins, in some segments of European population wine consuming can contribute up to 15% of daily OTA intake. Since wine is the second source of OTA, second only to cereals, we used wine as a representative of liquid food samples for practical experiments. In this study, we selected two common wines to verify the adsorption efficiency of the nano-Fe3O4@CTS for OTA. The results showed that the nano-Fe3O4@CTS had a good removal on contaminated wine, and the removal rate in white wine was 46.0%~63.3%, red wine was 68.4%~79.4. OTA contamination is also widespread in beer, juice and grains. Because the OTA removal approaches in liquid food samples through nano-Fe3O4@CTS is much the same. In our opinion, nano-Fe3O4@CTS can be applied to beer and juice. However, solid food samples, such as grains, need to be soaked and OTA immersed in solution before adsorption treatment. The soaking treatment may lead to some issues but this study does not make a further study. Therefore, we suggest the method mentioned in this research is more suitable for OTA removal in liquid samples.
- What is the long-term stability, reusability, and cost-effectiveness of nano-Fe3O4@CTS, particularly under repetitive adsorption-desorption cycles?
Response: Thank you for your professional consideration. We evaluated the stability and reusability of nano-Fe3O4@CTS by 4 adsorption-desorption. The results showed that after 3 adsorption-desorption, the removal rate increased from 69.4% to 63.5% (figure 1), and there was no significant change. The removal rate decreased to 59.9 % after the 4th cycle. Therefore, we consider that the nano-Fe3O4@CTS has good stability and reusability. In addition, the reusability of the adsorbent significantly reduces the demand for new adsorbents, thereby reducing initial investment and long-term costs, and has significant environmental benefits. On the other hand, other adsorbents such as activated carbon and macroporous resin adsorption are difficult to desorb, which leads to low reusability and affects economic efficiency. Therefore, the improvement of reusability is an important way to increase the economic benefits of adsorbents.
Figure 1. The reusability of nano-Fe3O4@CTS
- How do the size distribution, morphology, and synthesis methods of nano-Fe3O4@CTS influence adsorption performance, production cost, and scalability?
Response: The particle size distribution of the adsorbent directly affects its adsorption efficiency. Smaller particle size usually provides a larger specific surface area, thereby increasing the number of adsorption sites and improving the adsorption capacity. Nanoparticles usually have higher surface area and more active sites, so they have high adsorption capacity. The morphology of the adsorbent (such as spherical, flake, etc.) determines the specific surface area and pore structure. For example, the spherical structure exhibits higher adsorption capacity due to its regular shape and larger specific surface area. The synthesis method directly affects the production cost of the adsorbent, and also affects the pore structure and specific surface area of the adsorbent. The particle size distribution, morphology and synthesis method of the adsorbent have a significant impact on the adsorption performance, production cost and scalability. Smaller particle size and suitable morphology can improve the adsorption efficiency, but it is necessary to balance the production cost and process complexity.
- What are the optimal conditions (e.g., pH, temperature, adsorbent dosage) for OTA removal across different wine types and food matrices?
Response: We are sorry that we have not illustrated the above problem well. In order to verify the adsorption capacity of the nano-Fe3O4@CTS, we only selected classic wines, White wine (Alcohol: 12.5 %, Grape Variety: Chardonnay, type: dry, 750 mL) and red wine (Alcohol: 12.5 %, Grape Variety: Cabernet, type: dry, 750 mL). The optimal conditions for the OTA removal in other different wine types and food matrices can refer to the adsorption time of 30 min, adsorption temperature of 25 °C, adsorbent dosage of 1 g, and solution pH of 5. Thank you very much for your professional advice, we will use the nano-Fe3O4@CTS for OTA removal in different samples in the follow-up study, which will help to expand its application range.
- How does the adsorption process compare with existing mycotoxin removal methods (e.g., activated carbon) regarding efficiency, cost, safety, and environmental impact?
Response: Chitosan originates from chitin and which widely present in marine organisms such as shrimp, crabs, and shellfish. Its extraction process is relatively simple, fast, recyclable and cost-effective. It is reported that the production cost of chitosan is much lower than that of activated carbon, metal oxide and other adsorbents. The preparation cost of chitosan-based adsorbent is about 0.16 $/g, which is far lower than other adsorbents. Else, chitosan is a natural green material with high biocompatibility, degradability, non-toxicity and harmlessness, which meets the requirements of environmental protection. Due to the presence of amino and hydroxyl functional groups, chitosan can be recycled many times after physical or chemical modification, which further improves its economy and practicability.
- What are the regulatory and consumer safety implications of using nano-Fe3O4@CTS in food and beverage applications, particularly regarding residues, health effects, and compliance with food safety standards?
Response: Thank you very much for your careful consideration of our paper. The structure of the nano-Fe3O4@CTS is mainly composed of chitosan-coated Fe3O4 microspheres. Fe3O4 has superparamagnetic properties and can be quickly adsorbed to one side under an external magnetic field. Therefore, there is no excess residual Fe3O4 in the wine. In addition, chitosan is a natural polysaccharide with biodegradability and biocompatibility, and is easy to decompose in water. Chitosan is widely used in the fields of biomedicine and food. Therefore, the application of chitosan in food and beverage is safe.
- Are there potential competitive or synergistic interactions between OTA and other mycotoxins, contaminants, or wine components that might influence adsorption behavior?
Response: Thank you for your valuable and thoughtful comments. During the initial physical adsorption process, the nutrients in the wine may compete with the OTA in the pores. This will affect the adsorption capacity of the nano-Fe3O4@CTS, thereby reducing the adsorption efficiency. The content of other mycotoxins and pollutants in wine is extremely low, so there is no significant difference in adsorption capacity. However, during the chemical adsorption process, the -NH2 (CTS) and -COOH (OTA) and carboxyl groups interact with each other, thereby reducing the interference of other contaminants.
- Can nano-Fe3O4@CTS be further modified or combined with other materials to improve specificity, adsorption capacity, and versatility for removing OTA and other mycotoxins?
Response: Thank you for your advice. It is necessary to further modify the adsorbent. Amino (-NH2) and hydroxyl (-COOH) groups are common active sites on the surface of chitosan. -NH2 is one of its main active sites and has high reactivity. It can be chemically modified by quaternization, sulfonylation, esterification etc. to give chitosan new functions. Hydroxyl groups can be used for chemical modification such as phosphorylation and alkylation to improve their solubility and biological activity. In addition, magnetic chitosan can improve its specific adsorption and separation performance by combining with graphene oxide (GO), carbon nanotubes (CNT), TiO2, silica gel, cyclodextrin and other materials. Through modification and compounding with other materials, the specificity and adsorption capacity of nano-Fe3O4@CTS can be improved, so as to achieve efficient adsorption of OTA and other mycotoxins. Besides, the aptamer and antibody modified magnetic chitosan can selectively recognize OTA, thereby improving the specificity. However, it will can raise the cost substantially. Our research group has carried out related research work, which can greatly improve the specificity and selectivity of nano-Fe3O4@CTS.

Reviewer 2 Report
Comments and Suggestions for Authors
This manuscript entitled “Removal capacity and mechanism of modified chitosan for ochratoxin A based on rapid magnetic separation technology” intends to explore the detoxification activity of a modificated chitosan against OTA in standard solutions and artificially contaminated wine matrices. A preliminary characterization study of the nano composite synthesized was carried out. The results of the study may increase the number of available products to adsorption OTA and possibly other mycotoxins in food products, but it fails in one of its main objectives, which is its effective use in wines. The selection of a single dose for the experiments limits the observation of the final effects in the wines. In my opinion, the introduction lacks more content and suffers from a lack of depth that puts the work in context, and another weak point is the selection of analytical parameters to control the effects in the wines studied.
The following suggestions should be considered by the authors to improve this contribution:
- Line 24: please, to complete this sentence as: “…are Penicillium and Aspergillus genera [1]”.
- Line 25: to change “literatures” with “researches”.
- Line 29: the last regulation published by EU was Commission Regulation 2022/1370 amending Regulation (EC) No 1881/2006 as regards maximum levels of ochratoxin A in certain foodstuffs.
- Line 30: please, to change: “Kochman et al analyzed…” with “Kochman et al. [5] analyzed…”, and to correct similar issues in the text.
- Line 34: it is important to note that fungal infestation and OTA contamination is produced worldwide, and in different agricultural and animal commodities, in addition to cereals and wine.
- Lines 37 to 46: At this point, the authors should complete this paragraph with a description of the different physical, chemical, and biological methods to removal or detoxify OTA in food or in wine, providing significant literature reviews. For instance: Yang et al. (2024), Water, 16:24, 3620.
- Line 47: Chitosan should be deeply described in the text: origin, properties, authorized use in wine, dosage limits, etc. Besides, previously published studies of chitosan to reduce OTA in wines should be cited, for instance: Mine Kurtbay et al., J. Agric. Food Chem. 2008, 56, 2541‒2545.
- Line 69: This sentence “…and the practical application was studied through the changes of wine quality” should be modified. The chemical parameters studied in wine samples do not sufficiently represent the level of quality or the changes produced before and after treatments.
- Line 69: This sentence “…nanoparticles are cost-effective, safe, and easily adsorbent for the removal of mycotoxins from foods” should be deleted from this section.
- Line 73: The authors should provide complete information of the OTA standard used in the study.
- Line 82: The authors should provide more characteristics of the wine samples used (variety, basic chemical oenological parameters, etc).
- Line 88: please, to change “…at 25±1 ͦ C…” with “…at 25 ± 1 ºC…”, and replace through the text.
- Line 110: Which OTA concentration was used?
- Line 126: please, change “r/min” with “rpm”.
- Line 179: Why this “elevated” dose of nanoparticles (50 g/L) for the experiments in wine matrix was selected? Have the authors considered the impact of ferric ion on wine composition?
- Line 191: The caption “2.10. The effect of sorbents on wine quality” should be changed with “2.10. Wine analysis”.
- The authors should explain the reasons for the selection of these analytical parameters were chosen to control the experiments (color value, transmittance and TSS). The analysis of intensity in red wine is measured at 3 wavelengths (OD Abs420+520+620 nm). For better information consult the OIV analysis methods “Compendium of International Methods of Analysis”.
- Line 225: Please, change “The total acid content can be represented as tartaric acid” with “The total acidity was expressed as tartaric acid (g/L)”.
- Line 234: The acronyms “XRD” and “VSM” should be defined at the first appearance. Maybe, at 2.3. section.
- Line 291: At Figure 3B, at x axe caption the correct is “quantity/g” instead of “quality/g”? Figure captions are missing an indication if error bars are standard deviation or standard error of the mean, please check entire Figures.
- Line 330: Please, to specify concentration compound in this sentence.
- Line 358: Table 4 is erroneous. Please, to insert the correct Table in the text.
- Line 360: Please, change this caption “The effect nano Fe3O4@CTS on wine nutritional quality”, because it’s imprecision. Chemical parameters related with nutritional composition of the wines were not studied.
- Line 375: The term “translucency” is not frequent or standard in wine related studies, please to supply another.
- Lines 378-380: In my opinion, this sentence related with pasteurized palm liquid would be deleted. The influence of the use of chitosan in palm liquid cannot be compared with its influence on the chromatic characteristics of wines.
- Line 384: Table 5 should be modified in order to clarity. This Table should be sorted in the first column by the type of wine, and in the second column by before and after the adsorption treatment.
- Line 425: The conclusions should be completed with the main results (numerical) obtained in the study, v.g.: OTA removal rates.
- Line 427: At Conclusions section, the authors should discuss the impossibility of using it in wines under the conditions tested.
- Lines 441-542: The format of references should be standardized according to the requirement of FOODS journal.
Author Response
This manuscript entitled “Removal capacity and mechanism of modified chitosan for ochratoxin A based on rapid magnetic separation technology” intends to explore the detoxification activity of a modificated chitosan against OTA in standard solutions and artificially contaminated wine matrices. A preliminary characterization study of the nano composite synthesized was carried out. The results of the study may increase the number of available products to adsorption OTA and possibly other mycotoxins in food products, but it fails in one of its main objectives, which is its effective use in wines. The selection of a single dose for the experiments limits the observation of the final effects in the wines. In my opinion, the introduction lacks more content and suffers from a lack of depth that puts the work in context, and another weak point is the selection of analytical parameters to control the effects in the wines studied.
Response: Thank you for the above suggestions.
In this manuscript, firstly, we detected OTA in white and red wine stock solution, and the results were negative. Therefore, different volumes of OTA standard solution were added to the wine, and the contaminated wine was adsorbed by the nano-Fe3O4@CTS. The residual amount of OTA in the supernatant was determined and the removal rate was calculated. The results are shown in Table 5. The removal rate was 46.0 % ~ 63.3 % in white wine and 68.4 % ~ 79.4 % in red wine. In the follow-up study, we should further explore the effect of nano-Fe3O4@CTS in a variety of different types of wine, expand its application scope, and strive to achieve its application in wine.
Secondly, the selection of single parameters in this study were completed in OTA standard solution, which provided support for the kinetic and thermodynamic experiments and reduced the complexity of the model. In the initial exploration stage, due to the diversity of wine categories, we only chose two classical wine verification methods to verify the feasibility of the method. The choice of a single dose is suitable for the initial exploration stage, which can save time and resources and improve research efficiency. In order to overcome the limitations in wine, we should adopt multi-parameter optimization and uncertainty analysis to better adapt to complex sample.
Finally, we added some content in the introduction part, and cited previously published studied related to this work to enhance the depth of the manuscript.
We hope modified manuscript can meet your requirements.
Removal can effectively control the pollution of OTA in foods, thereby reducing the harm to humans and animals. At present, the known removal methods mainly include physical, chemical and biological methods. Chemical method can effectively re-move mycotoxin in foods through using strong oxidants, acids, alkalis and other chemicals, which convert OTA into other substances by chemical reaction and destroy the structure of mycotoxin [4]. Jalili et al [8] studied the removal of OTA in pepper by alkali ammonia, sodium bicarbonate, sodium hydroxide, potassium hydroxide and calcium hydroxide, respectively. The results showed that there was no significant differ-ence in the removal effect of the five compounds, and the highest sodium hydroxide was more than 50%. However, when OTA are removed by chemical methods, residual chemical reagents may have an impact on human health. In recent years, biological methods using microorganisms to remove and degrade mycotoxin have been identified as an effective and potential method. The mechanism is to destroy the chemical structure of the mycotoxin by microbiology, or the enzymes secreted by microbiology [4]. Nora et al. [9] found that commercial peroxidases could reduce OTA content by 41 % in grape juice within 5 h. During the production of wine and grape juice, Saccharomyces cerevisiae and lactic acid bacteria can effectively control the concentration of OTA [10]. Biological methods usually require specific microorganisms and enzymes, which often lead to higher production cost resources. In addition, since the efficiency of bio-logical methods may not be as high as that of physical or chemical methods, it requires longer time and greater investment. Physical adsorption methods manifest promising prospects in removing mycotoxins, such as heat treatment, radiation, adsorption, etc., but few of these physical removal materials have practical applications [11].
Chitosan is a natural polysaccharide mainly derived from chitin, and is obtained by partial deacetylation of chitin under alkaline conditions. The main sources of industrial scale production of chitosan are marine crustaceans, the shells of shrimps and crabs and the bone plates of squid [15]. Chitosan has gained popularity for its remark-able properties, such as innocuous, high adsorption and affinity, biocompatible, biode-gradable, antimicrobial, environment-friendly, and low cost [12]. In addition, the amino and hydroxyl groups in chitosan can be used as active sites and modified, mak-ing it widely used in different industries, such as pharmaceutical, food, agriculture, cosmetics, drug delivery, biotechnology, and biomedicine. Since 2011, the EU has ap-proved the use of chitosan for the removal of contaminants in wine, such as OTA. Bornet et al. [16] suggested chitosan as an adsorbent (2 and 5 g/L) can significantly reduce the concentration of OTA in red (56.7~83.4%), white (26.1~43.5%) and sweet wine (53.4~64.5%) The study also showed that chitosan can be used to clarify and eliminate OTA and other pollutants, such as metal ions. However, because of its viscosity, chitosan usually forms an emulsion in solution, which is difficult to separate in solution by traditional filtration, centrifugation, and sedimentation [17].
A novel magnetic graphene oxide modified with chitosan (MGO-CTS) was synthesized as an adsorbent aimed to removal mycotoxins OTA, ZEN, AFB1 and can be reduced to mycotoxins at 50 â—¦C and pH 5 [15].
- Line 24: please, to complete this sentence as: “…are Penicillium and Aspergillus genera [1]”.
Response: As suggested by the reviewer, the sentence “The principal producers of ochratoxin A (OTA) are Penicillium and Aspergillus.” was completed with “The principal producers of ochratoxin A (OTA) are Penicillium and Aspergillus genera.” (Line 24 of page 1)
- Line 25: to change “literatures” with “researches”.
Response: Thanks for your careful checks and it was rectified on Line 25 of page 1.
- Line 29: the last regulation published by EU was Commission Regulation 2022/1370 amending Regulation (EC) No 1881/2006 as regards maximum levels of ochratoxin A in certain foodstuffs.
Response: We feel sorry for our carelessness. We have already reviewed literature and replaced the “1881/2006” with “2022/1370” in the revised manuscript. (Line 29 of page 1)
- Line 30: please, to change: “Kochman et al analyzed…” with “Kochman et al. [5] analyzed…”, and to correct similar issues in the text.
Response: Thanks for your careful checks and it was rectified on Line 30 of page 1, and correct similar issues in the text.
- Line 34: it is important to note that fungal infestation and OTA contamination is produced worldwide, and in different agricultural and animal commodities, in addition to cereals and wine.
Response: Thank you very much for your advice, which is very helpful to our manuscript. We added the sentences “OTA has a wide range of contamination and can contaminate a variety of foods, such as corn, wheat, coffee, wine, juice, nuts and animal feed [6].” to explain in the text.
- Lines 37 to 46: At this point, the authors should complete this paragraph with a description of the different physical, chemical, and biological methods to removal or detoxify OTA in food or in wine, providing significant literature reviews.
Response: Thank you for your valuable and thoughtful comments. We have We have supplemented the explanation of OTA removal methods, including physical, chemical and biological methods, and provide significant literature review.
We wrote the following as a supplement in the introduction part:
Removal can effectively control the pollution of OTA in foods, thereby reducing the harm to humans and animals. At present, the known removal methods mainly include physical, chemical and biological methods. Chemical method can effectively re-move mycotoxin in foods through using strong oxidants, acids, alkalis and other chemicals, which convert OTA into other substances by chemical reaction and destroy the structure of mycotoxin [4]. Jalili et al [8] studied the removal of OTA in pepper by alkali ammonia, sodium bicarbonate, sodium hydroxide, potassium hydroxide and cal-cium hydroxide, respectively. The results showed that there was no significant differ-ence in the removal effect of the five compounds, and the highest sodium hydroxide was more than 50%. However, when OTA are removed by chemical methods, residual chemical reagents may have an impact on human health. In recent years, biological methods using microorganisms to remove and degrade mycotoxin have been identified as an effective and potential method. The mechanism is to destroy the chemical structure of the mycotoxin by microbiology, or the enzymes secreted by microbiology [4]. Nora et al. [9] found that commercial peroxidases could reduce OTA content by 41 % in grape juice within 5 h. During the production of wine and grape juice, Saccharomyces cerevisiae and lactic acid bacteria can effectively control the concentration of OTA [10]. Biological methods usually require specific microorganisms and enzymes, which often lead to higher production cost resources. In addition, since the efficiency of bio-logical methods may not be as high as that of physical or chemical methods, it requires longer time and greater investment. Physical adsorption methods manifest promising prospects in removing mycotoxins, such as heat treatment, radiation, adsorption, etc., but few of these physical removal materials have practical applications [11].
- Line 47: Chitosan should be deeply described in the text: origin, properties, authorized use in wine, dosage limits, etc. Besides, previously published studies of chitosan to reduce OTA in wines should be cited.
Response: Thank for your comments. In the submitted manuscript, we have described the origin, properties, and authorized use of chitosan, and cite previously published studies on the reduction of OTA in wines by chitosan.
We wrote the following as a supplement in the introduction part:
Chitosan is a natural polysaccharide mainly derived from chitin, and is obtained by partial deacetylation of chitin under alkaline conditions. The main sources of industrial scale production of chitosan are marine crustaceans, the shells of shrimps and crabs and the bone plates of squid [15]. Chitosan has gained popularity for its remark-able properties, such as innocuous, high adsorption and affinity, biocompatible, biode-gradable, antimicrobial, environment-friendly, and low cost [12]. In addition, the amino and hydroxyl groups in chitosan can be used as active sites and modified, mak-ing it widely used in different industries, such as pharmaceutical, food, agriculture, cosmetics, drug delivery, biotechnology, and biomedicine. Since 2011, the EU has ap-proved the use of chitosan for the removal of contaminants in wine, such as OTA. Bornet et al. [16] suggested chitosan as an adsorbent (2 and 5 g/L) can significantly reduce the concentration of OTA in red (56.7~83.4%), white (26.1~43.5%) and sweet wine (53.4~64.5%) The study also showed that chitosan can be used to clarify and eliminate OTA and other pollutants, such as metal ions. However, because of its viscosity, chitosan usually forms an emulsion in solution, which is difficult to separate in solution by traditional filtration, centrifugation, and sedimentation [17].
A novel magnetic graphene oxide modified with chitosan (MGO-CTS) was synthesized as an adsorbent aimed to removal mycotoxins OTA, ZEN, AFB1 and can be reduced to mycotoxins at 50 â—¦C and pH 5 [15].
- Line 69: This sentence “…and the practical application was studied through the changes of wine quality” should be modified. The chemical parameters studied in wine samples do not sufficiently represent the level of quality or the changes produced before and after treatments.
Response: As Reviewer suggested that we have modified the sentence “The nano-Fe3O4@CTS nanoparticle was applied to eliminate OTA in wine, and the practical application was studied through the changes of wine analysis.”
- Line 69: This sentence “…nanoparticles are cost-effective, safe, and easily adsorbent for the removal of mycotoxins from foods” should be deleted from this section.
Response: We sincerely thank the reviewer for careful reading. As suggested by the reviewer, we have deleted the sentence “The nano-Fe3O4@CTS nanoparticles are cost-effective, safe, and easily adsorbent for the removal of mycotoxins from foods”.
- Line 73: The authors should provide complete information of the OTA standard used in the study.
Response: As suggested by the reviewer, we have provided complete information of the OTA standard used in the study.
- Line 82: The authors should provide more characteristics of the wine samples used (variety, basic chemical oenological parameters, etc.).
Response: According to your suggestion. We have provided information of the wine, and the supplementary sentence is “White wine (Alcohol: 12.5 %, Grape Variety: Chardonnay, type: dry, 750 mL) and red wine (Alcohol: 12.5 %, Grape Variety: Cabernet, type: dry, 750 mL) both were gained from COFCO Great Wall Wine Co., Ltd. (Beijing, China).”
- Line 88: please, to change “…at 25±1 ͦ C…” with “…at 25 ± 1 ͦ C…”, and replace through the text.
Response: Thank you so much for your careful check. We have modified the mistakes in the new manuscript.
- Line 110: Which OTA concentration was used?
Response: According to your suggestion, we have now added OTA concentration (0.5 μg/mL) in the reviewed manuscript.
- Line 126: please, change “r/min” with “rpm”.
Response: According to your suggestion, we have now modified “r/min” to “rpm”.
- Line 179: Why this “elevated” dose of nanoparticles (50 g/L) for the experiments in wine matrix was selected? Have the authors considered the impact of ferric ion on wine composition?
Response: We are sorry for the vague understanding of this problem. According to our understanding, the nano-Fe3O4@CTS is the spherical structure formed by the distribution of Fe3O4 molecules in chitosan, and Fe3O4 usually exists in the form of nanoparticles rather than free Fe2+ or Fe3+. The role of Fe3O4 is to endow chitosan with magnetism, so that nano-Fe3O4@CTS can be quickly separated from the solution and improve the separation efficiency.
- Line 191: The caption “2.10. The effect of sorbents on wine quality” should be changed with “2.10. Wine analysis”.
Response: Thank you for good question. We have replaced “2.10. The effect of sorbents on wine quality” with “2.10. Wine analysis”.
- The authors should explain the reasons for the selection of these analytical parameters were chosen to control the experiments (color value, transmittance and TSS). The analysis of intensity in red wine is measured at 3 wavelengths (OD Abs420+520+620 nm). For better information consult the OIV analysis methods “Compendium of International Methods of Analysis”.
Response: We hope that our response will meet your requirements. In the study, we selected some indicators related to wine quality, rather than traditional wine quality parameters. Due to the pores on the surface of the nano-Fe3O4@CTS, some nutrients of wine will be intercepted during the adsorption process. In order to verify the feasibility of the method, we selected some parameters from the chemical aspects for determination. For example, the content of TSS is closely related to the alcohol content and preservation ability of wine. The color, brightness and turbidity of wine are closely related to its transmittance. By measuring the transmittance, wine can be better visually observed to assist tasting and quality assessment.
In addition, we are very sorry that the intensity analysis of wine at 3 wavelengths is not shown in the manuscript. Since the synthesis and characterization of the nano-Fe3O4@CTS takes a long time, we cannot supplement this part of the data in a short time.
Hope to get your understanding
- Line 225: Please, change “The total acid content can be represented as tartaric acid” with “The total acidity was expressed as tartaric acid (g/L)”.
Response: Thank for your comments. The sentence “The total acid content can be represented as tartaric acid” was replaced with “The total acidity was expressed as tartaric acid (g/L)”.
- Line 234: The acronyms “XRD” and “VSM” should be defined at the first appearance. Maybe, at 2.3. section.
Response: As for the reviewer’s advice, we have defined XRD and VSM in 2.3 section.
- Line 291: At Figure 3B, at x axe caption the correct is “quantity/g” instead of “quality/g”? Figure captions are missing an indication if error bars are standard deviation or standard error of the mean, please check entire Figures.
Response: Thank you so much for your careful check, we feel so sorry for our carelessness. We have corrected the X axis caption of Figure 3B. Else, we have now added figure captions in the reviewed manuscript. The sentence is “The error bar is expressed by standard deviation.”
Figure 3. (A) Effect of pH, (B) Effects of adsorbent dosage, (C) Effect of
adsorption temperature, (D) Effect of adsorption time. The error bar is expressed by standard deviation.
- Line 330: Please, to specify concentration compound in this sentence.
Response: According to your suggestion, we have modified it in the manuscript.
- Line 358: Table 4 is erroneous. Please, to insert the correct Table in the text.
Response: We are very sorry for the negligence of the Table 4. We re-inserted new tables 4 in the reviewed manuscript
- Line 360: Please, change this caption “The effect nano-Fe3O4@CTS on wine nutritional quality”, because it’s imprecision. Chemical parameters related with nutritional composition of the wines were not studied.
Response: Thank you for your professional advice, we have corrected “The effect nano-Fe3O4@CTS on wine quality “corrected to “Wine analysis”.
- Line 375: The term “translucency” is not frequent or standard in wine related studies, please to supply another.
Response: We sincerely thank the reviewer for careful reading. In the text, “translucency” was replaced by “Transmittance”.
- Lines 378-380: In my opinion, this sentence related with pasteurized palm liquid would be deleted. The influence of the use of chitosan in palm liquid cannot be compared with its influence on the chromatic characteristics of wines.
Response: Thank you for good question. We have deleted the sentence in new manuscript.
- Line 384: Table 5 should be modified in order to clarity. This Table should be sorted in the first column by the type of wine, and in the second column by before and after the adsorption treatment.
Response: According to your suggestion, we have modified Table 5 to clarity.
- Line 425: The conclusions should be completed with the main results (numerical) obtained in the study, v.g.: OTA removal rates.
Response: We are sorry for our negligence, and we have provided the main numerical results.
- Line 427: At Conclusions section, the authors should discuss the impossibility of using it in wines under the conditions tested.
Response: Thank you for pointing out this problem in manuscript. We wrote the following as a supplement in the conclusion part:
The nano-Fe3O4@CTS can effectively reduce the concentration of OTA, but OTA cannot be completely removed. Else, the concentration of OTA in wine may fluctuate with the fermentation process and storage conditions, so the efficiency of the nano-Fe3O4@CTS will also change. In future studies, nano-Fe3O4@CTS can be specifically modified to reduce its negative impact on wine, improve application range and removal efficiency.
- Lines 441-542: The format of references should be standardized according to the requirement of FOODS journal.
Response: Thank you very much for your suggestion. We have standardized the format of references according to FOODS journal.

Round 2
Reviewer 1 Report
Comments and Suggestions for Authors
Recommend accepting in the present form.
Reviewer 2 Report
Comments and Suggestions for Authors
The manuscript have been positively improved and it would be more likely to be accepted for publication. I have listed last corrections to do.
-Line 239: Please, change "Measurement of TSS" with "Measurement of total soluble solids".
-Line 258: Please, change "Measurement of total acid content" with "Measurement of total acidity content".
-Lines 463-465: This sentence: "Else, the concentration of OTA in wine may fluctuate with the fermentation process and storage conditions, so the efficiency of the nano-Fe3O4@CTS will also change", should be modified. The final OTA concentration in wines may fluctuate by fermentation process and treatments before bottling (clarification, filtration, etc).
-The references section has to be arranged using instructions for authors of Foods. Journal´s names must be in abbreviated form.
